# Exploring the Level of Post Traumatic Growth in Kidney Transplant Recipients via Network Analysis

**DOI:** 10.3390/jcm10204747

**Published:** 2021-10-16

**Authors:** Yuri Battaglia, Luigi Zerbinati, Martino Belvederi Murri, Michele Provenzano, Pasquale Esposito, Michele Andreucci, Alda Storari, Luigi Grassi

**Affiliations:** 1Nephrology and Dialysis Unit, St. Anna University Hospital, 44124 Ferrara, Italy; a.storari@ospfe.it; 2Department of Neuroscience and Rehabilitation, Institute of Psychiatry, University of Ferrara, 44124 Ferrara, Italy; zrblgu@unife.it (L.Z.); martino.belvederimurri@unife.it (M.B.M.); luigi.grassi@unife.it (L.G.); 3Nephrology and Dialysis Unit, Department of Health Sciences, Magna Graecia University, 88100 Catanzaro, Italy; michiprov@hotmail.it (M.P.); andreucci@unicz.it (M.A.); 4Department of Internal Medicine, Division of Nephrology, Dialysis and Transplantation, University of Genoa and IRCCS Ospedale Policlinico San Martino, 16132 Genoa, Italy; pasqualeesposito@hotmail.com

**Keywords:** post traumatic growth, psychiatric morbidity, kidney transplantation, network analysis, ESAS, MINI, CPC, DCPR, distress, demoralization, alexithymia, anxiety

## Abstract

Although kidney transplant can lead to psychiatric disorders, psychosocial syndromes and demoralization, a positive post-traumatic growth (PTG) can occur in kidney transplant recipients (KTRs). However, the PTG-Inventory (PTGI), a reliable tool to measure PTG is scarcely used to explore the effect of this stressful event in KTRs. Thus, the purpose of our study was to assess the level of PTG and its correlation with demoralization, physical and emotional symptoms or problems via network analysis in KTRs. Additionally, we aimed at exploring the association of PTG with psychiatric diagnoses, Diagnostic Criteria for Psychosomatic Research (DCPR) conditions, and medical variables. A total of 134 KTRs were tested using MINI International Neuropsychiatric Interview 6.0 (MINI 6.0), DCPR interview, PTGI, Edmonton Symptom Assessment System (ESAS), Canadian Problem Checklist (CPC) and Demoralization scale (DS-IT). PTGI was used to investigate the positive psychological experience of patients after KT. It consists of 21 items divided in five factors. Routine biochemistry, immunosuppressive agents, socio-demographic and clinical data were collected. A symptom network analysis was conducted among PTGI, ESAS and DS-IT. Mean score of PTGI total of sample was 52.81 ± 19.81 with higher scores in women (58.53 ± 21.57) than in men (50.04 ± 18.39) (*p* < 0.05). PTGI-Relating to Others (16.50 ± 7.99) sub-score was markedly higher than other PTGI factor sub-scores. KTRs with DCPR-alexithymia or International Classification of Diseases, tenth revision (ICD-10) anxiety disorders diagnosis had lower PTGI total score and higher PTGI-Personal Strength sub-score, respectively (*p* < 0.05). The network analysis identified two communities: PTGI and ESAS with DS-IT. DS-IT Disheartenment, DS-IT Hopelessness and PTGI Relating to Others were the most central items in the network. After 1000 bootstrap procedures, the Exploratory graph analysis revealed the presence of a median of two communities in the network in 97.5% of the bootstrap iterations. A more extensive use of PTGI should be encouraged to identify and enhance the positive psychological changes after KT.

## 1. Introduction

Kidney transplantation (KT) is the best treatment option for people affected by end stage chronic kidney disease (ESRD) [1]. However, KT can be considered a “traumatic event” for kidney transplant recipients (KTRs), leading to the development of intra-psychological conflicts and existential crisis [2]. Indeed, psychiatric disorders, especially anxiety and depression, are presented in KTRs with a prevalence from 10% to 40%, depending on the measures used, such as the International Classification of Diseases (ICD) or the Diagnostic and Statistical manual for Mental disorders (DSM) and cut-off points adopted [3,4,5].

Furthermore, other clinically significant psychosocial syndromes, not diagnosable with traditional psychiatric tools, were found in about 65% of KTRs. More specifically, abnormal illness behavior, irritability, alexithymia and somatization were the most frequent syndromes assessed by Diagnostic Criteria for Psychosomatic Research (DCPR), a useful interview for identifying sub-threshold or undetected syndromes [6]. 

Recently, demoralization, a psychological syndrome distinct from depression [7], has been diagnosed in a high percentage of medically ill patients [8], particularly with a high prevalence of up to 86% in KTRs [9]. That is a condition described as a state of existential distress, characterized by loss of meaning and purpose in life, hopelessness and helplessness and feelings of “being trapped” because of persistently being unable to cope with a particular stressful situation [10].

On the other hand, the stressful event of KT can promote positive psychological changes, which are at the basis of concept of post-traumatic growth (PTG). According to Tedeschi and Calhun [11], PTG is defined as a positive cognitive effect due to an extreme crisis, and not merely a coping mechanism in ordinary stress condition. Basically, PTG, induced by the activation of latent intrapersonal resources of the subject, might increase the level of mental functioning after a serious experience. In order to assess and screen the PTG, a specific psychometric tool, PTG Inventory (PTGI) has been developed [12]. It is able to capture five psychological dimensions: relating to others, new possibilities, personal strength, spiritual change and appreciation of life. Notably, although PTGI has been extensively used in traumatized healthy people [13], it has been also validated in several settings of medicine, such as psycho-oncology [14], neurology [15], hematopoietic stem cell [16] or liver transplantation [17] and dialysis [18]. Demoralization and PTG can be conceptualized as opposite reactions to a stressful event and, to date, little is known about the relationship between these two dimensions [9]. Thus, a network approach to psychopathology represents an intriguing option to investigate the associations between these two dimensions [19,20]. Rather than seeing mental disorders as the cause of different symptoms, this theory conceptualizes them as systems arising from the complex and mutual interaction of different symptoms [21,22]. This approach can be furtherly expanded with the use of Exploratory Graph Analysis (EGA) [23], allowing the identification of clusters of symptoms (“communities”) that are more related to each other. 

To date, only few studies [24,25] have been focused on the use of PTGI in KTRs; thus, the primary aim of our study was to evaluate the levels of PTG and to examine the relationship of PTG dimensions with demoralization, physical and emotional symptoms or problems via network analysis in KTRs. The secondary aim was to assess any association of the PTGI with psychiatric diagnoses, DCPR conditions and medical variables.

## 2. Materials and Methods

Ongoing KTRs, followed by the Nephrology Unit of the Ferrara University-Hospital, were enrolled in the study, during one of their routine follow up nephrological visits. Inclusion criteria were: (1) being a recipient of kidney from cadaveric or living donor and (2) age ≥ 18 years. Exclusion criteria were: (1) Karnofsky Performance Status Scale (KPS), indicating an insufficient level of autonomy (score < 50), and (2) presence of cognitive disorders (Mini Mental State Examination < 24). All patients gave their written informed consent, and the research protocol obtained the ethical approval from the Hospital Ethics Committee for Human Research (Code: 151297, on 17 March 2016). The procedures agreed with the Declaration of Helsinki. The same psychiatrist of the Consultation-Liaison Psychiatric Service, University Psychiatry Unit of the same Hospital tested each KTR. Two individual interviews, such as Mini-International Neuropsychiatric Interview (MINI6.0) and Diagnostic Criteria for Psychosomatic Research Interview (DCPR), as well as four self-report tools, including Post-Traumatic Growth Inventory (PTGI), Edmonton Symptom Assessment System (ESAS-Revised) and Canadian Problem Checklist (CPC) within the COMPASS (Comprehensive Problem and Symptom Screening) tool [26] and Italian version of Demoralization Scale (DS-IT), were administered.

### 2.1. PTGI

It is a screening measurement tool used to investigate the positive changes experienced in the aftermath of a traumatic event [27]. It consists of 21 items divided in five factors: New Possibilities, Relating to Others, Personal Strength, Spiritual Change and Appreciation of Life. Each item scored on 6-point Likert scale from 0 (I did not experience this change as a result of my crisis) to 5 (I experienced this change to a very great degree as a result of my crisis). A total score of posttraumatic growth was calculated [28].

### 2.2. Other Instruments

The Mini International Neuropsychiatric Interview (MINI 6.0) [29,30] is a diagnostic interview, expensively validated for DSM diagnoses and for ICD-10 diagnoses. In order to formulate a psychiatric diagnosis, the ICD-10 classification was used in this study.The Diagnostic Criteria for Psychosomatic Research Structured Interview (DCPR-SI) [31,32] is a semi-structured interview used to evaluate a set of 12 syndromes grouped in 3 different clusters: abnormal illness behavior (i.e., disease phobia, thanatophobia, health anxiety, illness denial); somatization and its different expressions (i.e., persistent somatization, functional somatic symptoms secondary to a psychiatric disorder, conversion symptoms, anniversary reaction); irritability (i.e., irritable mood and type A behavior), and in two clinical constructs (i.e., demoralization and alexithymia). It is able to identify psychological dimensions to a much greater extent than the DSM or ICD criteria [6,33].The Edmonton Symptom Assessment System (ESAS-Revised) [34,35] is a reliable assessment tool to explore the severity of six physical (i.e., pain, tiredness, nausea, drowsiness, lack of appetite, shortness of breath), three psychological (i.e., depression, anxiety, feeling of not being well) and one optional symptom (i.e., emotional distress corresponding to the Distress Thermometer) [36,37]. The Global Distress score (ESAS-TOTAL) was obtained by summing up all the scores on the single ESAS symptoms. Analogously, a physical distress sub-score (ESAS-PHYS) and a psychological distress sub-score (ESAS-EMOTIONAL) were computed as a sum of scores for the six physical symptoms and for the four psychological symptoms, respectively. The Italian version shows an acceptable level of validity and good psychometric properties in KTRs [38,39].The Canadian Problem Checklist (CPC) [40] is a useful instrument used to screen a list of 21 problems the patient has to deal with. It is divided into six categories (practical, social/family, emotional, spiritual, informational and physical problems) and the severity of each problem is rated in a yes/no (0–1) format.The Italian Version of the Demoralization Scale (DS-IT) [41] is a widely valid tool for measuring the demoralization in medical setting and also more recently in KTRs, over the past 2 weeks [9]. It consists in 24 items which compose four subscales: loss of meaning and purpose, dysphoria, disheartenment, and sense of failure. Each item ranges on 6-point Likert scale (0 = never; 5 = all the time) and the sum of the single subscales scores provides a total score.

Demographic data, clinical characteristics and routine biochemistry, determined using standard auto analyzer techniques, were collected.

### 2.3. Statistical Analysis

Categorical variables were shown as frequencies (%). Continuous variables were reported as either mean ± standard deviation (SD) or median and interquartile range (IQR) based on their distribution. T-test and chi-square test were employed to determine the associations of PTGI groups with DCPR/ICD-10 diagnosis, ESAS/CPC scores and clinical variables.

#### Exploratory Graph Analysis

Exploratory graph analysis (EGA) was used to estimate the number of dimensions in multivariate data using undirected network models [23]. EGA first applied a network estimation method followed by a community detection algorithm for weighted networks [42].

Network Estimation Method.

This study applied the graphical least absolute shrinkage and selection operator [43], which estimated a Gaussian Graphical Model [44] where nodes (circles) represented variables and edges (lines) represented the conditional dependence (or partial correlations) between nodes given all other nodes in the network. The ratio of the minimum and maximum λλ was set to 0.1 in order to control the sparsity of the network in the LASSO. EBICglasso was applied using the qgraph package in R and γγ was set to 0.5 [45]

Community Detection Algorithm.

The Walktrap algorithm was applied to detect the community and was implemented using the igraph package in R [46].

Data Analysis.

EGA was applied using the EGAnet package (version 0.9.8) in R (version 4.0.3) and its associated results were visualized using the GGally (version 2.1.1), ggplot2 (version 3.3.3), and qgraph (version 1.6.5) packages in R [47].

Bootstrap Exploratory Graph Analysis. 

Bootstrap exploratory graph analysis (bootEGA) was used to estimate and to evaluate the dimensional structure estimated using EGA. The parametric bootstrap procedure was implemented in this study. The procedure begun by estimating a network using EGA and then generating new replicate data from a multivariate normal distribution (with the same number of cases as the original data). EGA was then applied to the replicate data, continuing iteratively until the desired number of samples was achieved (e.g., 1000). The result was a sampling distribution of EGA networks. From this sampling distribution, descriptive statistics, namely median number of dimensions, 95% confidence intervals around the median and the number of times a certain number of dimensions replicates, were obtained. In addition, a median (or typical) network structure was estimated by computing the median value of each edge across the replicate networks, resulting in a single network. Such a network represented the “typical” network structure of the sampling distribution. The community detection algorithm was then applied, resulting in dimensions that would be expected for a typical network from the EGA sampling distribution. BootEGA was applied using the EGAnet package (version 0.9.8) in R (version 4.0.3) and its associated results were visualized using the GGally (version 2.1.1) and ggplot2 (version 3.3.3) packages in R [48].

## 3. Results

### 3.1. Characteristics of the Sample

In total, 134 consecutive KTRs were included in this study; nine patients declined to participate (six for work or family reasons and three because of health reasons) and none was excluded. Ninety patients were men, and the average age was 56.1 (SD = 12) years. The median time since transplantation was 85 months (ranged from 34 months to 178 months). Almost all patients (94%) received kidney from cadaveric donor and a short time in dialysis before KT (median = 22; IQR = 3–166 months) was found. More than half of KTRs (83) were on triple antirejection agents, including steroids (84.3%), calcineurin inhibitors (90.3%), mycophenolate (67.7%), mTOR inhibitors (8.3%), azathioprine (10.4%). Further detailed socio-demographic and clinical characteristics of the cohort are presented in Table 1.

### 3.2. Level of Post Traumatic Growth Inventory and Its Relationship with ICD Diagnoses and DCPR Diagnoses

The mean score of PTGI total was 52.81 (SD = 19.81). PTGI-Relating to Others (M = 16.50; SD = 7.99) sub-score was markedly higher than other PTGI factor sub-scores, including PTGI-New Possibilities (M = 11.42, SD = 5.43), PTGI-Personal Strength (M11.08, SD = 5.19), PTGI-Appreciation of Live (M = 9.92, SD = 3.55) and PTGI-Spiritual Changes (M = 4.33, SD = 2.91).

No significant differences of PTG total scores and sub-scores were found with socio-demographic (marital status, housing, occupation, kidney graft months, age), clinical characteristics (pre-emptive transplant, second transplant, immunosuppressive therapy, previous psychiatric ICD diagnosis and cause of nephropathy) and blood chemistry (hemoglobin, eGFR and creatinine), expect for sex and schooling. Women (M = 58.53, SD 21.57) had higher scores of PTGI than men (M = 50.04, SD 18.39) (*t* = 2.34, df = 130, *p* < 0.05) and a positive correlation was found between PTGI-New Possibilities and schooling (Pearson = 0.27, *p* < 0.05).

The specific characteristic of the DCPR and ICD diagnoses are extensively reported elsewhere [6], and briefly summarized in the Table 1.

Regarding ICD-10 psychiatric diagnosis (as assessed through the MINI6.0 interview), KTRs with anxiety ICD diagnosis had higher PTGI-Personal Strength sub-score (M 13.07 with SD 2.30) than ones without anxiety ICD diagnosis (M 10.84 with SD 5.39) (*t* = −2.81, df = 33.87, *p* < 0.05).

Regarding DCPR-diagnosis compared to the whole sample, both PTGI total score and all PTGI factors sub-scores was significantly lower in patients with alexithymia (M = 44.93, SD = 18.21, *t* = 2.53, df = 130, *p* < 0.05) and higher in nosophobic patients (M = 69.75, SD = 14.08, *t* = −1.99, *p* < 0.05).

### 3.3. Relationship of the PTG Level with DS-IT and COMPASS

The network of PTGI with DS-IT and ESAS is reported graphically in Figure 1, depicting the connection between physical and phycological symptoms [49].

In a first analysis, practical, social/family and informational problems from the CPC were included (emotional and physical problems were considered redundant in the model), but resulted in no connections and thus were not included in a second analysis. The most central items in the network were DS-IT Dishartenment, DS-IT Hopelessness and PTGI Relating to Others. The strength of each node as a measure of centrality is shown in Figure 2.

The values of edge weights in the network are reported in Table 2. After 1000 bootstrap procedures, the EGA revealed the presence of two communities in the network in 97.5% of the bootstrap iterations (95% CI 1.69–2.3).

## 4. Discussion

In this study, we report the level of Post Traumatic Growth by assessing PTG Inventory among KTRs and characterized, for the first time, the relationship of PTG with DS-IT, physical/emotional symptoms or problems via network analysis among patients who were submitted to kidney transplantation. Additionally, the effect of psychiatric and psychosocial variables on the development of PTG was identified.

A first result indicates a mean PTGI score of 52.81 (SD = 19.81). These findings are in line with the reported mean value of PTGI total score of a 3-year longitudinal study performed in 53 KTRs [50], supporting the evidence that posttraumatic growth often occur in patients after kidney transplant.

A plausible explanation could be due to the fact that KTRs are directly exposed to traumatic events, such as the transplant surgery, the relapse of previous kidney disease, the acute rejection, the infections and the side effects of immunosuppressive drugs, which can mobilize positive energy after kidney transplant. Besides, PTG might be also induced by the activation of other intrapersonal resources as consequence of a sense of gratitude towards the deceased donor and the medical team. Indeed, kidney transplantation is considered the beginning of a new life by KTRs, after a period of physical suffering for dialysis dependence and fear of remaining on dialysis for all life.

In depth, when investigating the different factors of PTGI, our data show PTGI-Relating to others sub-scale had the highest score among different dimensions of PTGI, putting in evidence another relevant aspect, the relationships of KTRs with their family. A strong familiar support could become crucial to overcome the extreme stress of transplant event, especially in presence of kidney living donor. These results are consistent with a recent study in which higher PTGI score in recipients from a living donor (M = 74, SD = 16) compared to from a cadaveric donor (M = 65, SD = 21) were reported, highlighting the influence of family members to enhance PTG in the post-transplant period [25]. On the other hand, we do not find correlation between the extent of post traumatic growth and months since transplantation. This finding seems to indicate that PTG is a phenomenon relatively stable overtime, similarly to other studies in patients with liver transplant [28], breast cancer [51] and colon-rectal cancer [52].

In relation to the network developed, the results reveal the presence of two distinct communities. More specifically, ESAS physical/emotional symptoms with DS-IT dimensions and PTGI factors were the first and second cluster, respectively. The symptom associations are not equally strong in the network and the edges between symptoms within each community were higher than edges between communities. DS-IT Appreciation of life, DS-IT Hopelessness and ESAS-Emotional were the three symptoms which moderately connected the two communities. These findings seem to indicate that KTRs with high burden of physiological symptoms, including distress, in the presence of low levels of hopelessness, are able to develop a higher level of appreciation of life after the kidney transplant. However, the relationship of distress with PTG is still debated in literature. Although many studies supported an inverse correlation between distress and PTG, the presence of distress symptoms could enhance or reduce the changes in the personality-related domains on the base of the time elapsed since traumatic event and the type of event (acute vs. chronic disease) [53].

Furthermore, our data seem to suggest that interventions aimed to promote appreciation of life could increase PTGI and decrease demoralization: interventions with a focus on valued-living are gaining importance, as the two dimensions might have possible common underlying processes; also, valued living has been shown to be correlated with PTGI-Appreciation of Life [54]. Other approaches might include those derived from positive psychology, like well-being therapy, which has been shown to increase PTGI in traumatized patients [55].

Other intriguing results emerged when the relationship of PTGI with ICD-10 was assessed. Paradoxically, the diagnosis of ICD-10 anxiety was positively associated with high PTGI-Personal Strength score. This result might be interpreted by the fact that higher self-awareness to handle difficulties is induced by anxiety. This cognitive state could be considered as a positive response of patients to the new organ’s integration [56]. However, this finding is in contrast with a previous report in which a persistent state of anxiety, due to the partial psychological integration of kidney, contributed to reduce the level of PTG. In addition, lower PTGI score was a significant predictor of graft rejection episodes at 3 years in cadaveric kidney transplant recipients (Odds Ratio = 0.963, 95% CI = 0.929–0.999, *p* < 0.05). In other words, the higher the psychological growing after KT, the lower rates of graft rejection [49].

On the other hand, it is worth noting that the presence of DCPR alexithymia, shown in about one-quarter of KTRs and in almost two-third of KTRs with another DCPR diagnosis [6], was a strong predictor of low PTGI score. Alexithymia is characterized by the inability to recognize and express own emotions [57]. This lack of symbolizing the feelings, a defense mechanism against a traumatic event as kidney transplant [58], could play a role in reducing the development of positive changes after KT. These data are consistent with the only other study, recently published, in which the levels of alexithymia, assessed by using TAS20 questionnaire [59], were negatively and moderately associated with PTGI total score and sub-scores, such as changes in self-perception and appreciation of life, in cadaveric donor recipients [50].

The strength of this study is that it evaluated, in KTRs, the level of the post-traumatic growth level as well as its correlations with DS-IT and emotional symptoms, using PTGI and network analysis, respectively. This research helps to promote the use of PTGI after the experience of kidney transplantation, as proved in heart, lung and liver transplantation [60,61,62]. Furthermore, the network analysis, a consolidated approach in psychopathology, represents an innovative technique in the kidney transplant research.

The limitations of our study are: (1) the small sample size of our population that does not allow us to generalize our results and (2) the lack of control group of patients with other kidney conditions. Further multicenter studies on larges cohort of KTRs could be conducted using a short form of PTGI [8,63,64], after its validation in the kidney transplant setting. Furthermore, (3) in our cohort the main cause of CKD was quite different from other investigated populations, however this variable seems not to be correlated with PTG score [24,25]. Another limit (4) is the use of the old version of DCPR-SI, due to the absence of the new version (DCRP-R) when the study started. In addition, although the administration of vitamin D is associated with an improvement of mental health [65,66], (5) we did not evaluate the effect of vitamin D on PTG. Additionally, (6) we did not consider the intensity of physical activity, a rising variable which can modify psychosocial and physical risks factors [67,68,69,70].

## 5. Conclusions

This study showed a moderate level of PTG in kidney transplant recipients, suggesting that health professionals responsible for treatment of KTRs should be sensitive to promote the awareness of these complex psychological changes after kidney transplant in their patients.

Furthermore, the network analysis identified two communities, ESAS physical/emotional symptoms with DS-IT dimensions and PTGI factors, which were more strongly connected within-cluster than between-clusters. This result suggests that a psychological intervention could be an appropriate means of addressing psychological distress, reducing hopelessness and improving the appreciation of life, among kidney transplant recipients. However, the connection between these three relevant dimensions, should be considered for further research to better define this relationship and its consequence in term of outcomes in KTRs.

## Figures and Tables

**Figure 1 jcm-10-04747-f001:**
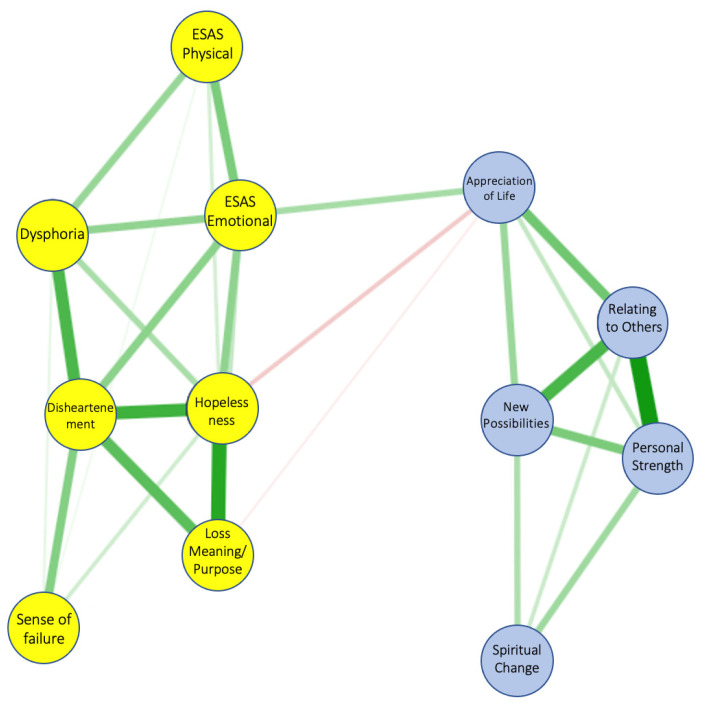
The network represents the relationships of PTGI factors with ESAS symptoms and DS-IT dimensions. Lines between symptoms (edges) are colored in green when they represent positive correlations and in red when they represent negative correlations. The magnitude of color represents the degree of the relationship between symptoms. ESAS: Edmonton Symptom Assessment System; ESAS-PHYS: physical distress sub-score; ESAS-Emotional: psychological distress sub-score.

**Figure 2 jcm-10-04747-f002:**
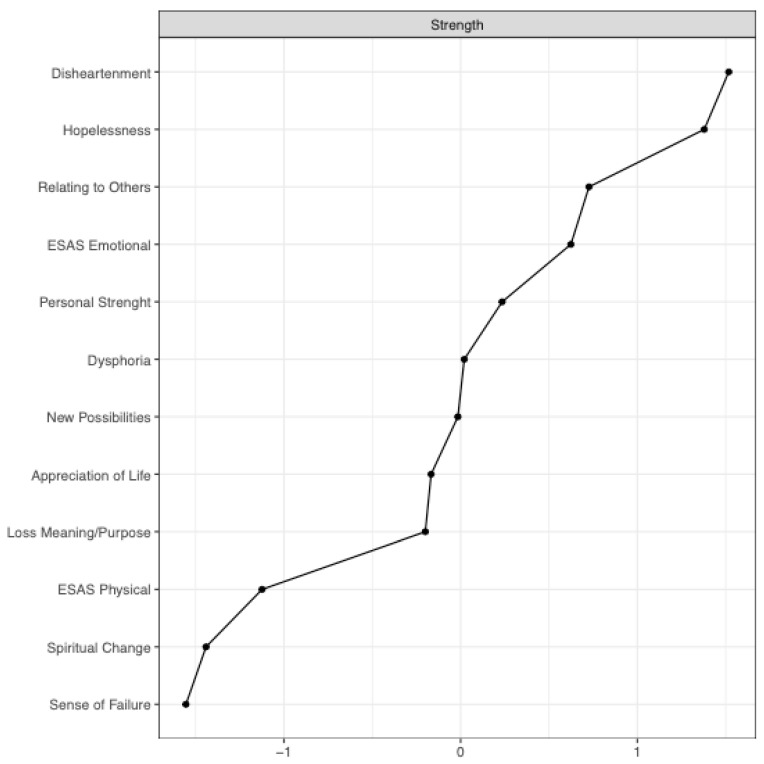
The centrality measure: the raw values of node strength are represented on horizontal axis, while the physical and phycological symptoms are represented on vertical axis.

**Table 1 jcm-10-04747-t001:** Socio-demographic characteristics, clinical variables and ranking order of DCPR and ICD diagnoses of the sample.

Variables		Variables	
Age, years	56.1 ± 12	*Occupation*	
Females, n (%)	44 (32.8)	Employed, *n* (%)	65 (48.5)
Race-Caucasian, *n* (%)	126 (94)	Retired, *n* (%)	59 (44.1)
Education, years	11.5 ± 4.52	Unemployed, *n* (%)	8 (6.0)
Smokers, *n* (%)	14 (10.4)	Unemployable, *n* (%)	2 (1.4)
Second Transplant, *n* (%)	10 (7.4)		
Acute Rejection, *n* (%)	12 (8.9)	*Marital Status*	
		Single, *n* (%)	29 (21.5)
*Cause of CKD*		Married, *n* (%)	89 (66)
Glomerulonephritis, *n* (%)	55 (41)	Divorced, *n* (%)	10 (7.5)
ADPKD, *n* (%)	25 (18.7)		
Diabetes Mellitus, *n* (%)	6 (4.5)	*DCPR Diagnosis*	
Hypertension, *n* (%)	4 (3)	Cluster AIB, *n* (%)	43 (31.3)
Other, *n* (%)	44 (32.8)	Cluster Irritability, *n* (%)	42 (31.3)
		Cluster Somatization, *n* (%)	26 (19.3)
*Living Situation*		Alexithymia, *n* (%)	31 (23.1)
Family, *n* (%)	93 (69.4)	Demoralization, *n* (%)	23 (17.2)
Parents, *n* (%)	22 (16.4)		
Alone, *n* (%)	11 (8.2)	*ICD-10 Diagnosis*	
Others, *n* (%)	8 (5.9)	No diagnosis, *n* (%)	88 (65.7)
		Anxiety disorders, *n* (%)	14 (10.4)
*Blood Test Values*		Mood [affective] disorders, *n* (%)	11 (8.2)
Creatinine serum, mg/dL	1.4 ± 0.54	Reaction to severe stress and adjustment disorders, *n* (%)	21 (15.7)
GFR-MDRD, mL/min	53.2 ± 17.5		

ADPKD: Autosomal Dominant Polycystic Kidney Disease; AIB: Abnormal Illness Behavior; BMI: Body Mass Index; CKD: Chronic Kidney Disease; DCPR: Diagnostic Criteria for Psychosomatic Research; ICD: International Classification of Diseases; GFR-MDRD: Glomerular Filtration Rate according to the equation from the Modification of Diet in Renal Disease Study.

**Table 2 jcm-10-04747-t002:** Edge weights, reflecting the connection strength of PTGI factors with ESAS Scores and DS-IT dimensions, in the network.

	ESAS-PHYS	ESAS-Emotional	Relating to Others *	New Possibilities *	Personal Strength *	Spiritual Change *	Appreciation of Life *	Loss of Meaning and Purpose **	Dysphoria **	Disheartenment **	Hopelessness **	Sense of Failure **
ESAS-PHYS		0.21							0.16		0.07	0.03
ESAS-Emotional	0.21						0.14	0.09	0.17	0.18	0.17	
Relating to Others *				0.29	0.40	0.08	0.22					
New Possibilities *			0.29		0.21	0.13	0.16					
Personal Strength *			0.40	0.21		0.15	0.09					
Spiritual Change *			0.08	0.13	0.15		0.01					
Appreciation of Life *		0.14	0.22	0.16	0.09	0.01		−0.03			−0.08	
Loss of Meaning and Purpose **		0.09					−0.03			0.26	0.34	
Dysphoria **	0.16	0.17								0.28	0.13	0.04
Disheartenment **		0.18						0.26	0.28		0.30	0.19
Hopelessness **	0.07	0.17					−0.08	0.34	0.13	0.30		0.08
Sense of Failure **	0.03								0.04	0.19	0.08	

* PTGI factors; ** DS-IT dimensions. The connections are highlighted in dark grey (strong) and light grey (moderate) for ease of visual comparison. ESAS: Edmonton Symptom Assessment System; ESAS-PHYS: physical distress sub-score; ESAS-EMOTIONAL: psychological distress sub-score; PTGI: Post Traumatic Growth Inventory.

## Data Availability

The data are not publicly available due to privacy restrictions.

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
