# Peer review of "Exploring the Level of Post Traumatic Growth in Kidney Transplant Recipients via Network Analysis"

_jcm, 2021, doi:10.3390/jcm10204747_

Round 1

Reviewer 1 Report

1) As a practicing nephrologist havent really thought of or heard from patients as kidney transplant process itself as a "traumatic' event so this was a different perspective for me. The days following with any complication could understandably be traumatic though.  As mentioned in the discussion the reason for trauma could be relapse of kidney disease or acute rejection, infections. In this study for what percent of patients had this kidney as a second transplant or had suffered a acute rejection episode needing treatment.

2) As mentioned there is physical suffering with dialysis and wondering if any studies comparing the PTG score between HD/PD and transplant patients could be cited ( if available) and also if there is a significant difference in the score between incenter and home HD populations.

3) Just a comment realized that most patients had GN with a minor set of population with diabetes and hypertension as cause of CKD. It seems to be quite different from the population in US. Wonder if that would affect the generalizability of the study to a different set of population who suffers more from metabolic syndrome complication.

Overall good study!

Reviewer 2 Report

The Authors presents a study aimed at exploring the level of post traumatic growth in kidney tranplant recipients (PTG) via a symptoms network analysis. No significant differences of PTG total score ans subscores were found with socio-demographyc, clinical characteristics chemistry. Women presented higher scores of PTGInventory then men and there was a significant positive correlation between PTGI-New Possibilities and schooling.  The AA conclude that the negative association found between PTGI score and DCPR-alexithymia suggests a routinary use of PTGI to identify patients in need of psychological support.

To the best of out knowledge tis is the first time such analysys has been carried out. However, the paper has several extremely weak points.

Major concerns:

-Small sample size: need for a multicenter study design;

-Lack of comparison between age classes, preemptive versus non preemptive kidney tranplantation, first or second transplant, number of comorbidity etc.

-No need for a case control (definetly not with patient still on the waiting list or other SOT ) but a better study design could include the evaluation before (at the time of  listing) and after transplantation and ideally after a few yeras.

We understand very well the difficulty of such a design study, but the paper in the present form does not add anything to the knowledge in the field and does not provide anything of practical use.

Minor concerns:

The paper is dificult to read. Its core part is somehow dismissed with the figure 1 that is supposed to be self explaining but it is not.

Reviewer 3 Report

MANUSCRIPT ID: jcm-1387077

General evaluation

This paper focuses on the concept of post-traumatic growth in kidney transplant recipients.

The aim of this study was to evaluate the levels of post-traumatic growth and to examine the relationship of post-traumatic growth dimensions with demoralization, physical and emotional symptoms or problems via network analysis in kidney transplant recipients. The secondary aim was to assess any association of the post-traumatic growth Index with psychiatric diagnoses, Diagnostic Criteria for Psychosomatic Research conditions, and medical variables.

This is a relevant topic for clinical practice, as posttraumatic growth often occur in patients after kidney transplant.

The correlation with the other scales are very interesting and the network analysis also.

The main recommendation was the early identification of psychological distress due to kidney transplant process.

Some comments to improve the Paper:

Please adapt figure 1, the text in the circle is too small.

In the Discussion you wrote about Vitamin D, why did you choose Vit D and not Anämia or Melatonin level or secondary effect of Antirejection therapie on Fatigue. Please explain the choice.

You suggest a multicenter studies on larges cohort of KTRs with a control group. Do you have any feasibility suggestions? A large cohort will never be able to send an extensive Questionnaire every year as you did in this small study. Which Items of the questionnaire would you suggest to integrate in a large cohort?

Round 2

Reviewer 2 Report

We thank the AA for their reply and comment.
